# Exploration of Disparities in Regions and Specialized Fields of Day Surgery System

**DOI:** 10.3390/ijerph17030936

**Published:** 2020-02-03

**Authors:** Beata Gavurova, Samer Khouri, Samuel Korony

**Affiliations:** 1Faculty of Mining, Ecology, Process Control and Geotechnologies of the Technical University of Košice, 040 01 Košice, Slovakia; samer.khouri@tuke.sk; 2Faculty of Economics, Matej Bel University, 974 01 Banska Bystrica, Slovakia; samuel.korony@umb.sk

**Keywords:** day surgery, correspondence analysis, regional disparities

## Abstract

The main objective of this paper was to find similarities among eight Slovak regions from the viewpoint of five specialized day surgery fields and among specialized day surgery fields from the viewpoint of Slovak regions on the basis of day surgery operated and hospitalized patient counts. Day surgery data of paediatric patients and of adult patients from the National Health Information Centre during the years 2009–2017 were used. Correspondence analysis in two dimensions of the Slovak regions and of specialized day surgery fields was applied in order to achieve the paper’s objective. The Košice Region differs most from the overall national average in both groups of paediatric day surgery. This is caused by its largest proportions in the fields of Gynaecology (29.7%) and Urology (48.0%) (operated patients), and in the fields of Gynaecology (60.5%) and Surgery (21.6%) (hospitalized patients). The most different specialized day surgery fields from overall average are: Urology (operated paediatric patients), Gynaecology (hospitalized paediatric patients), Otorhinolaryngology (operated adult patients) and Ophthalmology (hospitalized adult patients). Urogenital system day surgery procedures (Gynaecology, Urology) are separated from other three fields (i.e., Surgery, Ophthalmology, Otorhinolaryngology) either in the first or in the second dimension of the singular value matrix decomposition.

## 1. Introduction

Day surgery is the most effective form of health care provision as it ensures public resources savings. In Slovakia, this form of healthcare provision was regulated in 2004, and subsequently it has become standard practice. Its shares gradually increased from 6% at the beginning of its use to the current 30% of total surgical procedures. Currently, there are approximately 90 day surgery centers in various medical fields registered in Slovakia [1]. Despite the development of day surgeries, Slovakia still mostly uses hospital care. The biggest obstacle for patients is the number of supplements for one-day surgery. Slovak patients prefer to choose a hospital in which they do not pay for the surgery. Therefore, hospitalizations in inpatient healthcare facilities do not decrease in spite of the increasing number of surgeries performed in day surgery (operations without hospitalization). The United States and Canada maintain the primacy in the development of day surgery and the number of surgeries; among the Nordic countries in Europe, Sweden is the leading country. This difference is caused by an attitude of healthcare professionals, as well as health insurance companies, which prefer day surgery as far as possible. Social aspects also play an important role. There is no pressure from the patients themselves to minimize the number of hospitalizations, which is a serious obstacle to its further development. A positive signal in the global increase in surgeries of day surgery is the increase in the number of surgical procedures, supported by technical and technological progress, as well as the development in the anaesthesia field. The proportion of mini-invasive techniques in several fields of specialization has increased. In many countries, regional disparities in the number of day surgery operations are evident, with differences not only within geographically delimited sites (region, district), but also between hospitals in the same site. This high variability in performance between hospitals may reflect a lack of expertise of some anaesthesia teams in managing perioperative pain outside the hospital, as well as the experience of surgeons with particular surgical procedures, patient co-morbidity, etc. Few studies have been concerned with examining the variability among day surgeries from a regional perspective and revealing factors that may be significant in the insufficient progression of day surgery in a country [2,3,4]. The investigation of these facts is very important in setting relevant policies and regional strategic health plans, in designing health care expenditure in the context of demographic aging populations, and in assessing the effectiveness of the health system as a whole [5,6,7]. This was a motivation for this research. The aim of this study is to find out the relative positions of the Slovak regions in a five-dimensional space of day surgery fields and vice versa, positions of day surgery fields in an eight-dimensional space of the Slovak regions. Correspondence analysis in two dimensions was used. The aggregate count of both operated and hospitalized day surgery patients in two age groups (paediatric and adult patients) during the years 2009–2017 represent the starting point of this paper.

## 2. Overview of the Research Studies

Research studies that deal with day surgery belong to two fields of science: Medicine and medical management. Medical studies primarily focus on casuistry, and they provide little information for the economic sphere. Their aim is to improve diagnostic and medical treatment processes, while sharing medical experiences. Medical management studies have a broad research scope. Organization of the day surgery system, technical support, and economic and capacity parameters are their priority. Many studies from this group analyze patient dissatisfaction, as well as factors that influence their risk level [6,7,8,9]. These studies are significantly heterogenous in their character. However, they provide valuable topics for comparative analyses and research realizations in the individual conditions of the medical facilities. For instance, Anderson et al. [10] emphasize the importance of choosing patients correctly for a day surgery regime. Supporting mechanisms for the correct selection of patients and for the success of the operation are technical improvements in surgery and the use of surgical guidelines. A well-trained, motivated, multidisciplinary team in healthcare facilities where day surgery is considered is the gold standard. The authors state that the selection of patients for day surgery has to be precise, taking into account health and social factors, surgical constraints and other aspects that are part of the preoperative assessment. The importance of supportive mechanisms and guidelines for overnight surgery is also described in the study by Ng and Mercer-Jones [11]. In the UK, day surgery is constantly evolving and expanding. Competent institutions support the development of surgical and anaesthetic techniques as well as integrated health care processes, while using day surgery for a wider range of patients and using more complex procedures. The authors emphasize the importance of the right choice of patients and take into account specific factors that could influence the decision to undergo surgery in the mode of day surgery. In 2011, the Association of Anaesthetists of the United Kingdom and Ireland (AAGBI) issued guidelines for safe and effective day surgery, which are of great importance in preoperative and surgical processes. The goal of continuous improvement of the day-to-day surgery processes is to return a patient to normal life as quickly as possible. Leroy et al. [4] pointed out in their study the variability in the choice of day surgery interventions between Belgian hospitals in the years 2011–2013. The authors used administrative data analyses which were realized by surgical experts. In total, 486 surgical procedures were analyzed. Expert meetings with 95 doctors and anaesthesiologists from 54 hospitals were also used. The variability in the proportion of day surgery interventions between Belgian hospitals was considerable. This was also confirmed in standard performances with so-called high national daily surgical performance such as cataract surgery, laparoscopic cholecystectomy, etc. The study concluded that if Belgium expects a further expansion of daily surgery, factors that cause a high variability in the number of daily surgical interventions between hospitals should be further explored. A feedback system that would provide data on the evolution of the number of overnight surgeries between hospitals and other providers, and a benchmarking process for the monitoring of several quality indicators, such as unplanned re-hospitalizations and re-operations, visit of emergency medical units, etc., are very important. The high variability in the proportions of day surgery performance may be due to the experience and a sub-specialization of the surgical team. Lipp and Hernon [12] confronted the rate of day-to-day surgery in the UK with the procedures used. Over the past 20 years, the number of day surgeries has increased by more than five times as a result of improved surgical and anaesthetic techniques. Improvements in surgical experience have made it possible to use more complex surgeries with a higher rate of co-morbidity in patients. At the same time, an increasing rate of complications was not reported. The authors emphasize that day surgery should be a preferred option after considering the patient’s condition, planned surgical procedures and anaesthetics. Patients with multiple co-morbidities may be suitable for minor surgical procedures if general anaesthesia may be avoided. The authors also mention the importance of guidelines in establishing new units and the use of new procedures, the importance of the geographical location of units, and the experience in implementing more demanding surgical procedures. Brebbia et al. [13] analyzed the surgical department in multidisciplinary day surgery in their study. Between 2003 and 2006, the number of day surgeries increased considerably due to an organization of a good quality and a favorable population acceptance. The authors confirmed that the current organization of day surgery may maintain a high standard of health care and a low incidence of complications compared to the results of studies. The results show that day surgery in Italy is not evenly distributed and there are significant differences between regions. The authors see issues with the implementation of deeper analyses regarding the incompatibility of data resulting from the performance of interventions in the state and private health sector. The study highlights the importance of optimal linkages between organizational, management and clinical aspects. Ahmed et al. [14] deals with the infrastructure for the provision of day surgery. They are developing a building energy simulation model for a separate day surgery center. The detailed energy audit presented provides an insight into the occupancy rate and operation schedules. The issue of optimizing the cost of performing day surgery and finding opportunities for better organization, economic and personnel utilization also comes to the forefront. Gabriel et al. [15] discuss in more detail the risk factors and causalities associated with surgery and mortality. The results of the study did not confirm any association between perioperative mortality and the time of day at which operations were performed. Attention is also drawn to a determination of mortality rates for daily and emergency operations. The results of the analyses showed that the time of day was not associated with increased risk of mortality during intraoperative and immediate postoperative period in emergency surgery. Gaucher et al. [2] investigated the problem of stagnation in the development of overnight surgery in France during this period. Ambulatory surgery is restricted by law to “day surgery” in 12 h, and only 17 procedures are referenced for this surgery. According to the authors, the development of overnight surgery in France had progressed by that time (37% of all surgeries compared to 83% in the United States, 79% in the United Kingdom and 70% in Northern Europe). Conventional hospitalization remained the rule after surgery. In January 2010, university general surgery unit was restructured. It evolved from a conventional unit to a predominantly ambulatory unit. Research results show that it is clearly possible to distinguish the need for care of the need for accommodation and significantly reduce postoperative conventional accommodation. The authors call for a solution to the question of extending the legal period of 12 h to 24 h in order to expand the list of referenced procedures that could support the development of overnight surgery in the country. There are very few studies declaring the most appropriate models of the number and structure of medical staff for day surgery [5,6,15,16,17,18,19,20]. Experience in performing day surgery in many countries points to the necessity of its own administrative infrastructure in order to ensure coordinated planning and continuity in the performance of day surgery as well as overall optimization. Many foreign studies (e.g., [17,21,22,23,24,25,26,27,28], and others) have declared an effort to continually expand the day surgery system, both the number of patients operated and the application of more extensive and complex procedures (e.g., [7,8,9]) in patients with higher rates of comorbidity. This creates the preconditions for a more in-depth examination of regional disparities in the development of the number of surgeries performed and the factors that cause them. Their knowledge will help to set up regulatory and stabilization mechanisms in the country in order to achieve a higher efficiency of the health system and to ensure a better quality of life for patients. 

## 3. Materials and Methods 

The data for analysis were offered by the National Health Information Centre of Slovakia. The counts of operated patients and counts of hospitalized patients of both paediatric and adult day surgeries in the Slovak regions during 2009–2017 were analyzed. The Slovak regions with day surgery facilities were considered as the primary research objects. The data values are an aggregation of individual surgical procedures counts during 2009–2017 in Slovakia. The Official Gazette of the Ministry of Health of the Slovak Republic presents seven specialized fields of day surgery procedures: Surgery, Orthopaedics, Surgical Emergency and Plastic Surgery (hereinafter Surgery), Gynaecology and Obstetrics, Ophthalmology, Otorhinolaryngology, Urology, Dental Surgery and Gastroenterological Surgery and Gastroenterology. The day surgery procedures for the last two fields are presented to a lesser extent so they were not included in analyses.

In the paper, the correspondence analysis method was used [25]. The goal of correspondence analysis is a description of the relationship between two categorical variables of cross table in a low-dimensional space. However, it may be also used for relationships among categories of row or column variables separately. For each variable, the distances between category points in a plot reflect the relationships between the categories with similar categories plotted close to each other. Factor analysis or principal components are standard techniques for describing relationships between variables in a low-dimensional space. However, they require interval (continuous) data, and the number of observations should be at least five times the number of variables. Correspondence analysis, on the other hand, assumes nominal variables and may describe the relationships between categories of each variable, as well as the relationship between the variables. The graphical result of correspondence analysis is a so-called correspondence map, which depicts the relative positions of rows and columns of a cross table. Table rows (herein the Slovak regions) which are close together in a correspondence map are similar based on their proportions in table columns (herein specialized fields). Near table columns (Slovak day surgery specialization fields) are similar from the viewpoint of their proportions in table rows (regions). The cross-table has eight rows and five columns. From a geometrical viewpoint, the relative distances of eight Slovak regions in a five-dimensional space of specialized fields and the distances of five specialized fields in the eight-dimensional space of the Slovak regions are the objectives of analysis. Statistical software IBM SPSS version 25 was used for correspondence analysis with the following settings: chi-square distance, removed row and column means, canonical normalization, and biplot. Only basic formulas from SPSS manuals are presented. 

The first step was to compute auxiliary matrix Z with elements: (1)$$z_{ij}=\frac{f_{ij}}{\sqrt{f_{i+}f_{+j}}}-\frac{\sqrt{f_{i+}f_{+j}}}{N}$$
where
*f_ij_* = count value for row *i* and column *j*,*f_i+_* = row total marginal,*f_+j_* = column total marginal,*N* = total sum.

Then, singular value decomposition of *Z* was computed:*Z* = *KΛL’*(2)
where *K´K = I*, *L´L = I*, and *Λ* is a diagonal matrix of singular values. 

The arrays of both the left-hand singular vectors and the right-hand singular vectors were adjusted row-wise to form scores that are standardized in the row and in the column marginal proportions:(3)r˜is=kisfi+/N
(4)c˜js=ljsf+j/N

Then, both sets of scores satisfy the standardization restrictions simultaneously. Depending on the normalization option chosen, the scores are normalized, which implies a compensatory rescaling of the coordinate axes of the row scores and the column scores. The general formula for the weighted sum of squares that results from this rescaling is as follows:
for row scores:(5)$$\sum f_{i+}r_{is}^{2}=N\lambda_{s}\left(1+q\right)$$
for column scores:(6)$$\sum f_{+j}c_{js}^{2}=N\lambda_{s}\left(1-q\right)$$
where
*q* = 0 for canonical case,*q* = 1 for row principal case and*q* = −1 for column principal case.

Canonical normalization was used in the paper.

## 4. Results

Correspondence analysis begins with cross table of two categorical variables counts. There are four groups of day surgery patients: operated paediatric patients, hospitalized paediatric patients, operated adult patients and hospitalized adult patients. Official abbreviations of the National Health Information Centre of the Slovak regions (Banská Bystrica—BC, Bratislava—BL, Košice—KI, Nitra—NI, Prešov—PV, Trnava—TA, Trenčín—TC and Žilina—ZI) are used. For specialized fields of day surgery procedures, official medical abbreviations are used (Surgery, Orthopaedics, Surgical Emergency and Plastic Surgery—SURG, Gynaecology and Obstetrics—GYN, Ophthalmology—OPHTH, Otorhinolaryngology—ORL and Urology—UROL). 

### 4.1. Paediatric Day Surgery Results

The first group of analyzed day surgery patients involves operated paediatric patients (Table 1). The row variable is a group of eight Slovak regions and column variable are the five specialized day surgery fields. Together, there are 40 cells in the table. Any reliable conclusions from single counts are impossible (with the exception of trivial statements like “there is the largest count in case of Surgery in the Banská Bystrica Region”). This is a reason why correspondence analysis was developed—to depict cross table. 

The situation is not much better in the case of row proportions (e.g., the Banská Bystrica Region has the largest proportion in Otorhinolaryngology field (0.481) and the Trenčín Region has the smallest proportion in Ophthalmology (0.009), Table 2). In the case of uniform distributions, all row proportions should be equal to 0.2 (1/5). However, it is not possible to conclude which regions and fields are similar. 

The column proportions are presented in Table 3, from the viewpoint of each field’s contribution in the Slovak regions (e.g., Gynaecology has the largest relative proportion (29.7%) in the Košice Region). The expected uniform distribution column proportions are 0.125 (1/8).

Correspondence analysis was used to examine the similarity of the Slovak regions from the viewpoint of specialized fields of day surgery procedures and for the similarity of specialized fields according to the Slovak regions. The next report is the first important one in correspondence analysis: A dimension inertia summary (Table 4). Inertia is a measure of variance to explain the variance of the cross table with corresponding dimensions. The proportion of inertia in the first dimension (last but one column) is quite high (0.737). The second dimension below adds another 0.199 of inertia. The proportion of inertia explained by two dimensions (last column) is 93.6%, so two dimensions are enough for reliable conclusions about the relative positions of the Slovak regions and of specialization fields of day surgery in paediatric day surgery counts. The chi-square test is very large because of the large cross table counts (14,301.792; *p* < 0.001). The total inertia is 0.120 = chi-square test/total count of patients = 14,301.792/119,200.

The next report concerns point (region) contributions to inertia of dimension and vice versa, dimension contributions to inertia of points (regions) (Table 5). If the contributions of regions are distributed equally to the inertia, then the contributions would be 0.125 (1/8). The Košice Region contributes a substantial portion (0.603) to the inertia of the first dimension. On the other side, the Banská Bystrica Region contributes mainly to the inertia of the second dimension (0.540). The Prešov Region contributes to both dimensions (0.223, 0.304). The contributions of the other five Slovak regions to inertia are relatively small in both dimensions. Two dimensions contribute to a large amount of the inertia for most row (regional) points. Relatively large contributions of the first dimension to regions of Košice (0.959), Bratislava (0.862) and Žilina (0.883) indicate that these regions are well-represented in one dimension. Higher dimensions contribute less to the inertia of these regions. This means that they are close to the depicted first dimension. The second dimension contributes mainly to the Banská Bystrica Region (0.965). Its first coordinate is zero, so it is right on the second dimension. Nitra (0.222) is explained poorly by two dimensions (see column Total in Table 5). The last three regions (Prešov, Trnava and Trenčín) are roughly equally explained by the first two dimensions.

The field of Urology contributes mainly (0.812) to the first dimension, while Surgery explains a large amount (0.724) of the inertia for the second dimension. Other fields contribute less to either dimension. In two dimensions, all fields with exception of Ophthalmology (total = 0.091) are well-represented. Fields of Otorhinolaryngology (0.716) and Urology (0.991) are represented well in the first dimension. The second dimension particularly contributes to Surgery (0.979). The fields of Gynaecology and Ophthalmology are not well-represented in two dimensions. The graphical presentation of correspondence analysis results enables visual cluster analysis of rows (regions) and columns (fields). In Figure 1, there are positions of the Slovak regions (red full circles) according to operated paediatric patient counts from the viewpoint of their proportions in specialized fields of day surgery and vice versa, day surgery fields (blue full circles) from the viewpoint of their proportions in the Slovak regions. The Košice and the Prešov Regions are adjacent. On the other side, regions of Bratislava, Trenčín and Žilina make one group, meaning they have similar proportions in the specialized fields of day surgery. Regions of Košice and Žilina are well explained by the first dimension, while Banská Bystrica is well explained by the second dimension (Nitra is not well-represented in two dimensions). The first dimension divides the Slovak regions into two groups— Košice on the right side vs. the rest of Slovakia on the left side. The second dimension divides the regions of Eastern Slovakia (below first dimension) vs. the rest of Slovakia (above first dimension). The Košice Region is near to the fields of Urology and Gynaecology. Thus, there is a possible association between the Košice Region and these fields. The proportions of the Košice Region are the largest ones from all of the Slovak regions in the fields of Urology (48.0%) and Gynaecology (29.7%). Similarly, Prešov is near to Otorhinolaryngology with the largest proportion (24.6%) in that field and Banská Bystrica to Surgery (21.4%). 

However, it is different in the case of the specialized day surgery fields. All fields are isolated so they are not similar in their proportions in the Slovak regions. The first dimension divides fields into two groups—urogenital fields (Gynaecology and Urology) vs. other fields. The second dimension separates the Surgery field vs. all remaining fields. Urology is well explained by the first dimension, Surgery by the second dimension. Urology is the most distant field from overall average, while Otorhinolaryngology is the least distant field from average.

Results for the other three groups of day surgery patients are similar. Therefore, they are presented more briefly. In the next part, results for the group of operated paediatric day surgery patients are presented (see Table 6). 

The report of inertia according to the dimensions is in Table 7. The proportion of inertia in the first dimension is large (0.707). The second dimension adds another 0.155 of inertia. The overall proportion of inertia explained by two dimensions is 86.1%. Two dimensions are enough for reliable conclusions about relative positions of the Slovak regions and of specialization fields for hospitalized paediatric day surgery patients. The chi-square test is very large (4520.489; *p* < 0.001). The total inertia is 0.275 = chi-square test/total count of patients = 4520.489/16,423.

The corresponding contributions of regions and fields in the group of hospitalized paediatric patients are shown in Table 8. The Košice Region contributes by a substantial portion (0.597) to the inertia of the first dimension. The contribution to the second dimension mainly comes from the Žilina Region (0.428). Other above average contributions (0.125) are from two regions: Trenčín in the first dimension (0.214) and Prešov in the second dimension (0.241). Contributions of the other five Slovak regions to inertia are relatively small in either dimension. Relatively large contributions of the first dimension are in regions of Košice (0.935), Nitra (0.638) and Trenčín (0.868). These regions are well-represented in the first dimension. The second dimension contributes mainly to Prešov (0.813) and Žilina (0.614). The regions of Banská Bystrica (total = 0.444) and Trnava (0.353) are explained poorly by two dimensions. 

For field contributions, the field of Otorhinolaryngology (0.355) contribute most to the first dimension, while Urology explains a large amount (0.313) of the inertia for the second dimension. The first dimension contributes well in the fields of Surgery (0.784), Gynaecology (0.780) and Otorhinolaryngology (0.953), while the second dimension contributes well to Urology (0.482). In two dimensions, all fields bar Urology are well-represented. 

In Figure 2, the positions of the Slovak regions and of day surgery specialized fields according to hospitalized paediatric patients are shown. Košice is again adjacent with the largest proportions in the fields of Surgery (21.6%) and Gynaecology (60.5%). The regions of Košice and Trenčín are well explained by the first dimension, while Prešov is well explained by the second dimension. The closest region to the overall average is Nitra. All day surgery fields are isolated so there are not similar profiles in their proportions in the Slovak regions. The second dimension separates Surgery from other fields. Otorhinolaryngology is well explained by the first dimension, and Urology by the second dimension. Gynaecology is the furthest field from the average profile.

### 4.2. Adult Day Surgery Results

The second part of the results is concerned with adult day surgery patients. In Table 9, there are counts of operated adult patients. 

In the case of operated adult patients, two dimensions are enough (Table 10). The proportion of inertia in the first dimension is 0.587. The second dimension explains another 0.240 of inertia. The proportion of inertia explained by two dimensions is 82.7%. Naturally, the chi-square test is larger, due to a higher number of counts (104,446.056; *p* < 0.001). The total inertia is 0.080 = (104,446.056/130,6781). 

The contributions of regions and fields to inertia are shown in Table 11. In the first dimension, an above average contribution (0.125) comes from two regions: Bratislava (0.449) and Prešov (0.489). In the second dimension, most inertia comes from Trenčín (0.482). Relatively large contributions of the first dimension are in regions of Bratislava, Prešov and Žilina. They are well-represented in the first dimension. In the second dimension, the contribution is large from the regions of Banská Bystrica and Trenčín. Two regions are not well-represented in two dimensions (Košice and Trnava).

With regards the field contributions, the fields of Ophthalmology (0.422) and Otorhinolaryngology (0.503) contribute most to the first dimension, while the field of Surgery (0.609) explains a large amount of the inertia for the second dimension. The largest contribution of the first dimension is in case of Ophthalmology (0.831). The second dimension contributes well to Surgery (0.891). Two fields are not well-represented (Gynaecology and Urology). 

The Slovak regions and day surgery specialized fields on the basis of operated adult patients are presented in Figure 3. The Bratislava, Prešov and Trenčín regions are adjacent. Bratislava has the largest proportion in the field of Ophthalmology (27.9%), Prešov in the field of Otorhinolaryngology (31.1%), and Trenčín in Surgery (17.1%). Nitra and Žilina are well explained by the first dimension, while Banská Bystrica and Trenčín are well explained by the second dimension. The closest region to the overall average is Žilina. 

Only three fields are explained well by two dimensions (Surgery, Ophthalmology and Otorhinolaryngology). The Otorhinolaryngology field has clear a position. The closest to the average profile is Gynaecology. The last group of day surgery patients in analyses are hospitalized adult patients. Counts are presented in Table 12. 

The approximation by two dimensions in the case of hospitalized adult patients is better (Table 13). The proportion of inertia in the first dimension is 0.573. In the second dimension, it is 0.261. Together, the proportion of inertia explained by two dimensions is 83.5%. The chi-square test is significant (26,317.098; *p* < 0.001) and total inertia is 0.166 = (26,317.098/ 158,461). 

The corresponding contributions of regions and fields are presented in Table 14. Three regions are above average in contribution to the first dimension: Košice (0.399), Prešov (0.140) and Žilina (0.155). In the case of the second dimension, only Bratislava (0.658) is above the average contribution. Relatively large contributions of the first dimension are in all regions, with the exceptions of Bratislava and Trnava. In the second dimension, a contribution is large in case of Bratislava (0.743). All regions are well-represented in two dimensions.

The fields of Surgery (0.318) and Urology (0.309) contribute most to the first dimension, while Ophthalmology (0.646) explains a large amount of the inertia for the second dimension. The largest contribution of the first dimension is from Surgery (0.903), Gynaecology (0.875) and Urology (0.663). The second dimension contributes well to Ophthalmology (0.687). Otorhinolaryngology is not well-represented in two dimensions. 

In Figure 4, the Slovak regions and specialized fields from the viewpoint of hospitalized adult patients are presented. Bratislava is adjacent to the center of Figure 4, with the largest proportion in the field of Ophthalmology (47.4%). The Košice region is near to the fields of Gynaecology (29.8%) and Urology (50.9%), with the largest proportions in them. Similarly, Trenčín is close to the fields of Surgery (18.1%) and Otorhinolaryngology (24.7%), which have the largest proportions. The nearest regions to the overall average are Trenčín and Banská Bystrica. Ophthalmology is clearly distant from other fields. The closest to average profile is the field of Surgery. The first dimension is almost exclusively defined by Surgery and Gynaecology. There is a polarity defined by the second dimension urogenital procedures (Gynaecology and Urology) vs. the other procedures (Otorhinolaryngology is not well depicted in two dimensions).

## 5. Discussion

There were few similarities among eight Slovak regions from the perspective of five specialized fields, and among the groups of paediatric operated and hospitalized patients, and operated and hospitalized adult patients, after applying correspondence analysis. In Slovakia, a realization of day surgery performances is not evenly distributed. Previous studies by Gavurova and Korony [6] and Soltes and Gavurova [29,30] reveal the disparities in the Slovak regions in terms of performance numbers and structure. Similar regional disparities were found in the study by Brebbia et al. [13], who analyzed the Italian regions based on a database from 2003–2006. The analysis results provided in the Table 15 show a summary of extreme values within individual regions. This information represents a platform for researching a day surgery development which depends on socio-demographic indicators in the individual regions, as well as on the parameters of a regional health policy, that is, access to healthcare, medical devices and equipment of a particular region, density of specialized medical facilities in the regions, etc. 

Other determinants that influence a development of day surgery in Slovakia may be revealed by researching causal relations between provided components. The given findings correspond with results of the study by Leroy et al. [4], who emphasized a high level of performance variability among hospitals. The authors also highlighted a significance of a feedback that should provide relevant data of day surgery performances in order to realize a benchmarking and a monitoring of multiple quality indicators. Similarly, in the Slovak conditions, the data provision is a result of the data structure in the day surgery databases which limits a realization of more structuralized analyses. On the other hand, return rates of day surgery annual statistical survey statements are over 90%, so there are no problems concerning representativeness and reliability.

New concepts and reforms in the Slovak health policy influence a development of the day surgery system from a macroeconomic point of view. Integrated healthcare and the stratification of hospitals is also considered here. The concept of integrated healthcare is implemented in the Strategic framework for health for 2014–2030, and its main aim is to increase a concentration of provided primary healthcare in Slovakia. The integrated healthcare centers (IHC) should have been created within this framework. Their aim was to make a healthcare more available for people, who live in those areas, where there is a lack of doctors, and consequently, patients are forced to commute to more distant regions in order to access health care services. These IHCs should have integrated general practitioners with specialists, with a possibility of providing social services at one place. In 2018, the Ministry of Health of the Slovak Republic introduced a concept of the stratification of hospitals whose aim is to significantly improve inpatient care by 2030. At present, there are significant differences in the quality of provided healthcare in the Slovak hospitals. It is also obvious in various levels of mortality, re-operation rates, re-hospitalization rates, etc. This project is based on the fact that the quality of healthcare performance increases with their number within a particular health facility. Consequently, the Slovak hospitals should be allowed to provide a specific specialization only in a case of achieving a required minimum number of performances. Also, an entire hospital network should be more efficient, with no negative impacts on patients in the regions. Regional centralization of some processes should provide a sufficient amount of resources. The transformation of almost 5600 beds and a subsequent release of many spaces and personal capacities, medical and operation sources could be achieved by more efficient processes in the hospitals. This is an opportunity for the optimal use of day surgery, long-term healthcare, and/or outpatient care [29,30]. Thus, it is necessary to use these sources efficiently and search for new possibilities in order to develop day surgery in the Slovak regions more rapidly with regards the development of a demographic ageing process that is connected to an increase of older population [31,32,33]. The results of analysis show similarities and differences of individual regions and specialized fields depending on a number of realized performance measures and a number of hospitalizations. Moreover, they provide a valuable platform for subsequent comparative analyses that would reveal the main reasons for different profiles of the individual regions and would identify obstacles of its insufficient progress. It is necessary to examine whether the reasons of insufficient progress in day surgery development in Slovakia lies in an incorrect health policy or in the changes of demographic structure, health behaviour, or in a different development of morbidity and mortality structures in the regions, etc. These aspects will be the subject for future research, and they require access to more structured data.

## 6. Conclusions

Day surgery should be the preferred option to hospitalized healthcare after considering a patient’s condition, the planned operation procedure and anaesthetic requirements. Many studies and a medical practice proved that the home environment has positive effects on a patient’s medical treatment and convalescence after surgery. Despite this fact, most of the regular non-complicated surgeries end in a long stay in a hospital. In a day surgery regime, a patient spends only the necessary time in a health facility. The length of stay in a hospital is shorter for a patient, and consequently, the number of treated patients increases and waiting times for surgical performances decreases. A patient’s quality of life and their arrival to a work–life balance is also very important. The development of day surgery in Slovakia is uneven. It is influenced by many factors. Regional analyses that would reveal disparities and discrepancies in the day surgery development, as well as its structure, are necessary for their identification and research. This represents a motivation for the research. 

The main objective of the paper was to find the relative positions of the Slovak regions from the point of specialized day surgery fields and of specialized day surgery fields from the perspective of the Slovak regions. Correspondence analysis was selected to achieve this. Correspondence analysis is an exploratory data analysis method. Its results may help disentangle the main aspects and patterns of cross table data. The region of Košice is the differs most from the overall national average in both groups of paediatric day surgery. This is caused by its largest proportions in the fields of Gynaecology and Urology (group of operated paediatric day surgery patients) and in the fields of Surgery and Gynaecology (group of hospitalized paediatric day surgery patients). Another adjacent region is Prešov (the largest proportion in Otorhinolaryngology in a group of operated paediatric day surgery patients). In the group of operated adult patients there are three adjacent regions: Bratislava (the largest proportion being in the field of Ophthalmology), Prešov (field of Otorhinolaryngology) and Trenčín (field of Surgery). The group of hospitalized adult patients has one clear outlying region, Bratislava, with its largest proportion in the field of Ophthalmology. The second outlying region is Košice (Urology and Gynaecology). On the other hand, the most similar regions to the average of Slovakia are Trenčín (operated paediatric patients and hospitalized adult patients), Prešov (hospitalized paediatric patients) and Žilina (operated adult patients). According to singular value decomposition, Eastern Slovakia is separated from the rest of Slovakia in a group of operated paediatric patients. The regions of Bratislava and Košice vs. other regions are divided in the group of hospitalized adult patients. The most different fields from the overall average are Urology (operated paediatric patients), Gynaecology ((hospitalized paediatric patients), Otorhinolaryngology (operated adult patients) and Ophthalmology (hospitalized adult patients). The most similar to overall average are the fields of Otorhinolaryngology (in both groups of paediatric patients) and Surgery (in both groups of adult patients). Urogenital system procedures (Gynaecology and Urology) are separated from the other three fields (Surgery, Ophthalmology and Otorhinolaryngology), either in the first or second dimension of the singular value matrix decomposition. The results form a valuable platform for the creators of health policies and regional, strategic health plans and concepts. Progress of day surgery development in Slovakia requires a continual realization of regional analyses that reveal disparities and discrepancies in terms of development. However, it also requires research of present pricing and regulatory mechanisms of health insurance companies, and economic and treatment processes in the hospitals.

## Figures and Tables

**Figure 1 ijerph-17-00936-f001:**
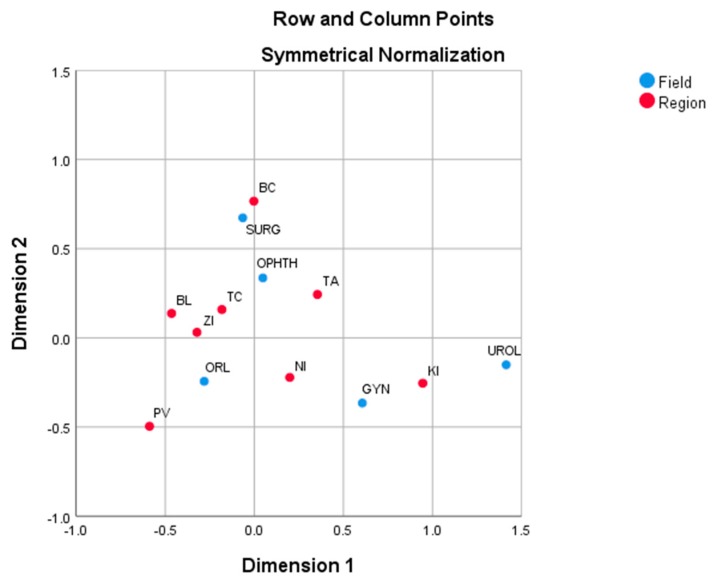
Correspondence plot of the Slovak regions and day surgery specialized fields on the basis of operated paediatric patients. Source: own analysis

**Figure 2 ijerph-17-00936-f002:**
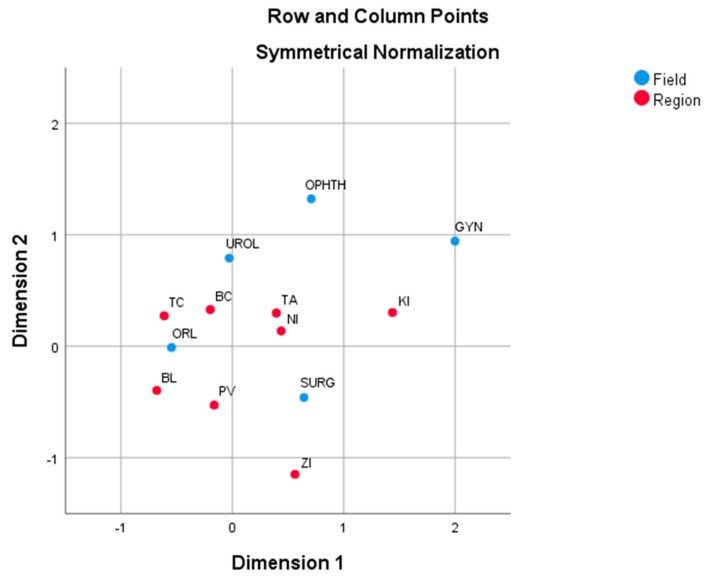
Correspondence plot of the Slovak regions and day surgery specialized fields on the basis of hospitalized paediatric patients. Source: own analysis.

**Figure 3 ijerph-17-00936-f003:**
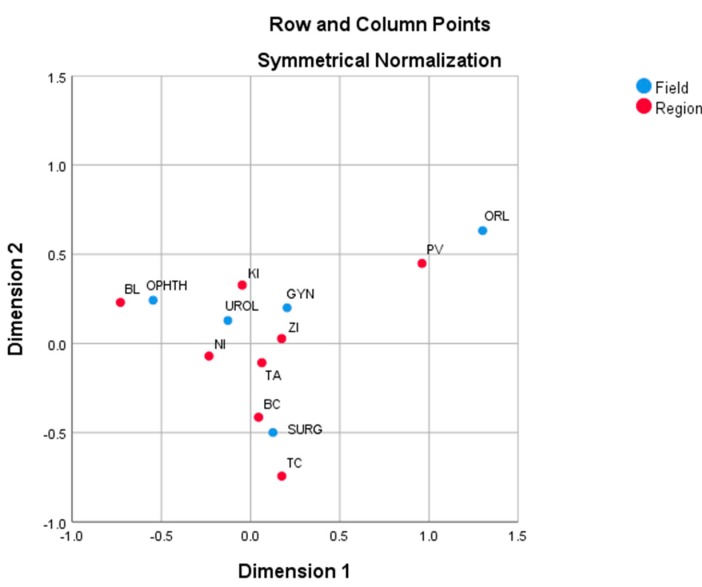
Correspondence plot of the Slovak regions and day surgery specialized fields on the basis of operated adult patients. Source: own analysis.

**Figure 4 ijerph-17-00936-f004:**
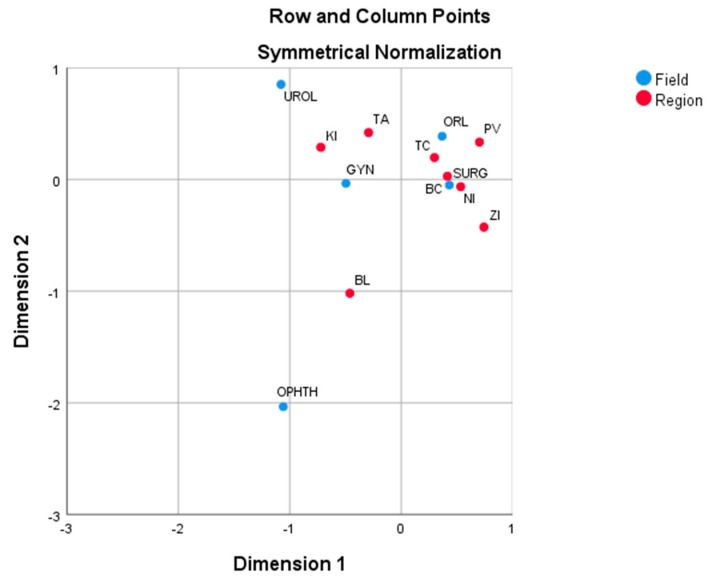
Correspondence plot of the Slovak regions and fields on the basis of hospitalized adult patients. Source: own analysis.

**Table 1 ijerph-17-00936-t001:** Cross table of day surgery operated paediatric patient counts.

Region	Field					
	SURG	GYN	OPHTH	ORL	UROL	Active Margin
BL	3473	42	122	8262	564	12,463
BC	6318	347	386	8146	1740	16,937
KI	4483	728	400	11,418	6911	23,940
NI	1462	404	48	3859	941	6714
PV	3963	393	279	17,499	665	22,799
TA	2372	119	1	4183	1499	8174
TC	2936	136	97	6394	943	10,506
ZI	4468	285	385	11,389	1140	17,667
Active Margin	29,475	2454	1718	71,150	14,403	119,200

Legend: Region: Banská Bystrica—BC, Bratislava—BL, Košice—KI, Nitra—NI, Prešov—PV, Trnava—TA, Trenčín—TC and Žilina—ZI; Specialisation field: Surgery, Orthopaedics, Surgical Emergency and Plastic Surgery—SURG, Gynaecology and Obstetrics—GYN, Ophthalmology—OPHTH, Otorhinolaryngology—ORL and Urology—UROL.

**Table 2 ijerph-17-00936-t002:** Row proportions of operated paediatric patient counts.

Region	Field					
	SURG	GYN	OPHTH	ORL	UROL	Active Margin
BL	0.279	0.003	0.010	0.663	0.045	1.000
BC	0.373	0.020	0.023	0.481	0.103	1.000
KI	0.187	0.030	0.017	0.477	0.289	1.000
NI	0.218	0.060	0.007	0.575	0.140	1.000
PV	0.174	0.017	0.012	0.768	0.029	1.000
TA	0.290	0.015	0.000	0.512	0.183	1.000
TC	0.279	0.013	0.009	0.609	0.090	1.000
ZI	0.253	0.016	0.022	0.645	0.065	1.000
Mass	0.247	0.021	0.014	0.597	0.121	

**Table 3 ijerph-17-00936-t003:** Column proportions of operated paediatric patient counts.

Region	Field					
	SURG	GYN	OPHTH	ORL	UROL	Mass
BL	0.118	0.017	0.071	0.116	0.039	0.105
BC	0.214	0.141	0.225	0.114	0.121	0.142
KI	0.152	0.297	0.233	0.160	0.480	0.201
NI	0.050	0.165	0.028	0.054	0.065	0.056
PV	0.134	0.160	0.162	0.246	0.046	0.191
TA	0.080	0.048	0.001	0.059	0.104	0.069
TC	0.100	0.055	0.056	0.090	0.065	0.088
ZI	0.152	0.116	0.224	0.160	0.079	0.148
Active Margin	1.000	1.000	1.000	1.000	1.000	

**Table 4 ijerph-17-00936-t004:** Inertia per dimension summary of operated paediatric patients.

Dimension	Singular Value	Inertia	Chi Square	Sig.	Proportion of Inertia	
					Accounted for	Cumulative
1	0.297	0.088			0.737	0.737
2	0.155	0.024			0.199	0.936
3	0.071	0.005			0.042	0.978
4	0.051	0.003			0.022	1.000
Total		0.120	14,301.792	<0.001	1.000	1.000

**Table 5 ijerph-17-00936-t005:** Contributions of regions and fields in a group of operated paediatric patients.

Region/Field	Mass	Score in Dimension		Inertia	Contribution				
		1	2		Of Point to Inertia of Dimension		Of Dimension to Inertia of Point		
					1	2	1	2	Total
BL	0.105	−0.465	0.137	0.008	0.076	0.013	0.862	0.039	0.901
BC	0.142	−0.003	0.766	0.013	0.000	0.540	0.000	0.965	0.965
KI	0.201	0.945	−0.254	0.056	0.603	0.084	0.959	0.036	0.995
NI	0.056	0.199	−0.222	0.005	0.007	0.018	0.135	0.087	0.222
PV	0.191	−0.588	−0.496	0.027	0.223	0.304	0.730	0.270	1.000
TA	0.069	0.354	0.243	0.005	0.029	0.026	0.549	0.135	0.683
TC	0.088	−0.182	0.159	0.002	0.010	0.014	0.577	0.228	0.805
ZI	0.148	−0.322	0.031	0.005	0.052	0.001	0.883	0.004	0.887
SURG	0.247	−0.065	0.673	0.018	0.004	0.724	0.017	0.979	0.996
GYN	0.021	0.606	−0.366	0.007	0.025	0.018	0.306	0.058	0.364
OPHTH	0.014	0.048	0.336	0.003	0.000	0.011	0.003	0.087	0.091
ORL	0.597	−0.281	−0.244	0.020	0.159	0.229	0.716	0.280	0.996
UROL	0.121	1.413	−0.151	0.072	0.812	0.018	0.991	0.006	0.997
Active Total	1.000			0.120	1.000	1.000			

**Table 6 ijerph-17-00936-t006:** Cross table of day surgery hospitalized paediatric patient counts.

Region	Field					
	SURG	GYN	OPHTH	ORL	UROL	Active Margin
BL	203	7	6	863	0	1079
BC	844	11	165	1745	506	3271
KI	1103	356	137	340	157	2093
NI	378	49	28	323	152	930
PV	1024	31	0	1694	178	2927
TA	330	60	0	312	194	896
TC	555	58	74	2976	465	4128
ZI	677	16	4	359	43	1099
Active Margin	5114	588	414	8612	1695	16,423

**Table 7 ijerph-17-00936-t007:** Inertia per dimension summary of hospitalized paediatric patients.

Dimension	Singular Value	Inertia	Chi Square	Sig.	Proportion of Inertia	
					Accounted for	Cumulative
1	0.441	0.195			0.707	0.707
2	0.206	0.043			0.155	0.861
3	0.168	0.028			0.103	0.964
4	0.099	0.010			0.036	1.000
Total		0.275	4520.489	<0.001	1.000	1.000

**Table 8 ijerph-17-00936-t008:** Contributions of regions and fields in a group of hospitalized paediatric patients.

Region/Field	Mass	Score in Dimension		Inertia	Contribution				
		1	2		Of Point to Inertia of Dimension		Of Dimension to Inertia of Point		
					1	2	1	2	Total
BL	0.066	−0.679	−0.396	0.022	0.069	0.050	0.605	0.096	0.701
BC	0.199	−0.199	0.329	0.018	0.018	0.105	0.195	0.249	0.444
KI	0.127	1.438	0.303	0.124	0.597	0.057	0.935	0.019	0.955
NI	0.057	0.438	0.137	0.008	0.025	0.005	0.638	0.029	0.667
PV	0.178	−0.163	−0.528	0.013	0.011	0.241	0.166	0.813	0.979
TA	0.055	0.395	0.298	0.013	0.019	0.023	0.279	0.074	0.353
TC	0.251	−0.612	0.274	0.048	0.214	0.091	0.868	0.081	0.950
ZI	0.067	0.562	−1.149	0.030	0.048	0.428	0.314	0.614	0.928
SURG	0.311	0.643	−0.459	0.072	0.292	0.318	0.784	0.187	0.972
GYN	0.036	1.999	0.943	0.081	0.324	0.154	0.780	0.081	0.862
OPHTH	0.025	0.708	1.322	0.022	0.029	0.214	0.254	0.415	0.669
ORL	0.524	−0.547	−0.011	0.072	0.355	0.000	0.953	0.000	0.953
UROL	0.103	−0.028	0.791	0.028	0.000	0.313	0.001	0.482	0.484
Active Total	1.000			0.275	1.000	1.000			

**Table 9 ijerph-17-00936-t009:** Cross table of day surgery operated adult patient counts.

Region	Field					
	SURG	GYN	OPHTH	ORL	UROL	Active Margin
BL	66,756	41,734	111,831	6193	12,238	238,752
BC	60,416	36,909	38,830	6402	6200	148,757
KI	59,338	63,389	68,738	13,030	15,841	220,336
NI	41,408	31,256	41,012	4077	5431	123,184
PV	47,047	41,164	30,179	26,193	5384	149,967
TA	35,765	27,405	25,487	5116	10,850	104,623
TC	75,702	28,113	37,546	9349	7077	157,787
ZI	56,332	37,449	46,615	13,893	9086	163,375
Active Margin	442,764	307,419	400,238	84,253	72,107	1,306,781

**Table 10 ijerph-17-00936-t010:** Inertia per dimension summary of operated adult patients.

Dimension	Singular Value	Inertia	Chi Square	Sig.	Proportion of Inertia	
					Accounted for	Cumulative
1	0.217	0.047			0.587	0.587
2	0.139	0.019			0.240	0.827
3	0.105	0.011			0.138	0.965
4	0.053	0.003			0.035	1.000
Total		0.080	104,446.056	<0.001	1.000	1.000

**Table 11 ijerph-17-00936-t011:** Contributions of regions and fields in a group of operated adult patients.

Region/Field	Mass	Score in Dimension		Inertia	Contribution				
		1	2		Of Point to Inertia of Dimension		Of Dimension to Inertia of Point		
					1	2	1	2	Total
BL	0.183	−0.729	0.230	0.025	0.449	0.070	0.854	0.054	0.908
BC	0.114	0.044	−0.414	0.004	0.001	0.141	0.014	0.760	0.773
KI	0.169	−0.047	0.327	0.005	0.002	0.130	0.015	0.470	0.485
NI	0.094	−0.233	−0.071	0.002	0.024	0.003	0.553	0.032	0.586
PV	0.115	0.961	0.449	0.027	0.489	0.167	0.835	0.116	0.951
TA	0.080	0.063	−0.108	0.005	0.001	0.007	0.014	0.026	0.040
TC	0.121	0.175	−0.743	0.011	0.017	0.482	0.073	0.850	0.923
ZI	0.125	0.174	0.027	0.001	0.017	0.001	0.793	0.013	0.806
SURG	0.339	0.125	−0.499	0.013	0.025	0.609	0.088	0.891	0.978
GYN	0.235	0.205	0.200	0.008	0.045	0.068	0.277	0.169	0.445
OPHTH	0.306	−0.547	0.242	0.024	0.422	0.130	0.831	0.104	0.935
ORL	0.064	1.301	0.632	0.029	0.503	0.186	0.806	0.122	0.928
UROL	0.055	−0.127	0.129	0.006	0.004	0.007	0.033	0.021	0.054
Active Total	1.000			0.080	1.000	1.000			

**Table 12 ijerph-17-00936-t012:** Cross table of day surgery hospitalized adult patient counts.

Region	Field					
	SURG	GYN	OPHTH	ORL	UROL	Active Margin
BL	8826	7461	2440	1264	888	20,879
BC	6927	3139	198	1809	282	12,355
KI	13,784	12,917	1586	2583	6626	37,496
NI	11,145	3527	119	720	514	16,025
PV	8784	2011	6	2483	633	13,917
TA	8154	5885	40	805	2220	17,104
TC	15,119	6653	254	3321	1569	26,916
ZI	10,823	1711	505	453	277	13,769
Active Margin	83,562	43,304	5148	13,438	13,009	158,461

Source: own analysis.

**Table 13 ijerph-17-00936-t013:** Inertia per dimension summary of hospitalized adult patients.

Dimension	Singular Value	Inertia	Chi Square	Sig.	Proportion of Inertia	
					Accounted for	Cumulative
1	0.309	0.095			0.573	0.573
2	0.208	0.043			0.261	0.835
3	0.148	0.022			0.132	0.966
4	0.075	0.006			0.034	1.000
Total		0.166	26,317.098	0.000	1.000	1.000

Source: own analysis.

**Table 14 ijerph-17-00936-t014:** Contributions of regions and fields in a group of hospitalized adult patients.

Region	Mass	Score in Dimension		Inertia	Contribution				
		1	2		Of Point to Inertia of Dimension		Of Dimension to Inertia of Point		
					1	2	1	2	Total
BL	0.132	−0.461	−1.020	0.038	0.091	0.658	0.225	0.743	0.968
BC	0.078	0.414	0.028	0.008	0.043	0.000	0.533	0.002	0.535
KI	0.237	−0.721	0.288	0.043	0.399	0.094	0.881	0.095	0.976
NI	0.101	0.534	−0.065	0.013	0.093	0.002	0.664	0.007	0.670
PV	0.088	0.701	0.334	0.020	0.140	0.047	0.653	0.100	0.753
TA	0.108	−0.293	0.419	0.010	0.030	0.091	0.276	0.383	0.659
TC	0.170	0.298	0.196	0.008	0.049	0.031	0.602	0.175	0.777
ZI	0.087	0.742	−0.427	0.025	0.155	0.076	0.591	0.132	0.723
SURG	0.527	0.431	−0.051	0.033	0.318	0.006	0.903	0.008	0.912
GYN	0.273	−0.496	−0.036	0.024	0.218	0.002	0.875	0.003	0.878
OPHTH	0.032	−1.058	−2.035	0.041	0.118	0.646	0.275	0.687	0.962
ORL	0.085	0.367	0.386	0.024	0.037	0.061	0.149	0.112	0.260
UROL	0.082	−1.078	0.851	0.044	0.309	0.285	0.663	0.279	0.942
Active Total	1.000			0.166	1.000	1.000			

**Table 15 ijerph-17-00936-t015:** Extreme values in the individual regions of operated and hospitalized patients.

Group	Region	Field (prop.)	Field (prop.)	Region	Field (prop.)	Region	Field (prop.)
Operated paediatric patients	KI	GYN (29.7%)	UROL (48.0%)	PV	ORL (24.6%)	BC	SURG (21.4%)
Hospitalized paediatric patients	KI	GYN (60.5%)	SURG (21.6%)	-	-	-	-
Operated adult patients	PV	ORL (31.1%)	-	BL	OPHTH (27.9%)	TC	SURG (17.1%)
Hospitalized adult patients	KI	GYN (29.8%)	UROL (50.9%)	BL	OPHTH (47.4%)	-	-

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
