# Peer review of "Exploration of Disparities in Regions and Specialized Fields of Day Surgery System"

_ijerph, 2020, doi:10.3390/ijerph17030936_

Round 1

Reviewer 1 Report

The aim of the paper is to find relative positions of the Slovak regions from the point of specialized day surgery fields and of specialized day surgery fields from the point of the Slovak regions.

Main comments

1. The topic of the paper is of great interest, especially in Central and Eastern Europe, where the efficiency of health expenditures is a major concern. See, in this regard and in order to highlight better the importance of the topic, Anton Sorin Gabriel, Onofrei Mihaela, ”Health Care Performance and Health Financing Systems in Countries from Central and Eastern Europe”, Transylvanian Review of Administrative Sciences, No. 35 E/2012, pp. 22-32

2. The paper is well structured and written.

3. The title of the article is clear and adequate.

4. The abstract is clear, it presents the object of research the content and the results.

5. The introduction states the objectives of the paper and the authors prove to know well the extant literature.

6. The methodology seems sound.

7. The results and interpretations are correct and refer to the results of previous studies.

Author Response

Dear reviewer

Thank you very much for your valuable comments. We are glad that the paper sounds well in your opinion.

Reviewer 2 Report

Comments on IJERPH manuscript

The authors set lofty goals:

“to find similarities among eight Slovak regions from 13 the viewpoint of five specialized day surgery fields and among specialized day surgery fields from 14 the viewpoint of Slovak regions” “Few research studies are concerned with examining the variability in day surgery from a regional perspective and revealing factors that may be significant in the insufficient progression of day surgery in a country [2, 3, 4]. Investigation of these facts is very important in setting relevant policies and regional strategic health plans, in designing health care expenditure in the context of 56 demographic aging populations, and in assessing the effectiveness of the health system as a whole [5, 6, 7].”

Unfortunately, their results focus more on differences rather than on similarities. 

And most importantly, I can find no discussion of the specific policy implications of the differences found, other than the need for additional studies.  General discussions at the top of page 17 about differences between the surgical volumes go as well with the simple cell counts as with the correspondence analysis. 

To this referee who is not a scholar of day surgeries, their review of the literature seems adequate.  But it is presented as one paragraph, despite several logical breaks in the arguments. 

While I am quite a fan of graphical representations of results, I was quite disappointed by the use of the neutral terms, Dimension 1 and Dimension 2 as axes labels.  The authors need to explain what the close clustering and/or distance from the horizontal or vertical axes mean, and discuss what differences have what policy significance.

Line 215:  “Any reliable conclusions from single cell counts are impossible.”

Line 217 and 218:  “This is a reason why correspondence analysis was developed – to depict cross table.” is an inadequate rebuttal to value of Table 1. 

Lines 223-227:  Similarly, this paragraph should be explaining what differences from the 1/5th row proportions contain more important information than the cell counts. 

Line 242-243: “Inertia is measure of variance to explain the variance of cross table with corresponding dimensions.”  Please explain more clearly.  The use of dimensions appears to have a similar roll to either factors in Factor Analysis or to Eigen values in Eigen Vectors.   

Throughout the paper, the authors distinguish between operated and hospitalized patients.  But the review of the literature would suggest that the authors are distinguishing between day surgery patients (ie those surgical patients not admitted to stay overnight or longer) versus between surgical patients that are hospitalized.  I would suggest “day surgeries” versus “hospitalized surgeries”. 

Author Response

Dear reviewer

Thank you very much for your valuable comments. Here you are, your suggestions are applied in this way:

1."The authors set lofty goals: “to find similarities among eight Slovak regions from 13 the viewpoint of five specialized day surgery fields and among specialized day surgery fields from 14 the viewpoint of Slovak regions” "

Correspondence analysis is typical exploratory data analysis. Any exploratory analysis method intends to uncover basic patterns or structures in data matrix (cases and (or) variables). Correspondence analysis does not emphasize statistical tests, though it is suitable to test for possible associations between analysed variables before its application. Our goals correspond to purpose of exploratory data analysis.

2."“Few research studies are concerned with examining the variability in day surgery from a regional perspective and revealing factors that may be significant in the insufficient progression of day surgery in a country [2, 3, 4]. Investigation of these facts is very important in setting relevant policies and regional strategic health plans, in designing health care expenditure in the context of 56 demographic aging populations, and in assessing the effectiveness of the health system as a whole [5, 6, 7].” Unfortunately, their results focus more on differences rather than on similarities."

Correspondence analysis is based on geometric viewpoint so distances among regions (fields) represent their relative (mutual) similarities. The larger (smaller) distances among regions (fields) mean the smaller (larger) similarities.

3."And most importantly, I can find no discussion of the specific policy implications of the differences found, other than the need for additional studies.  General discussions at the top of page 17 about differences between the surgical volumes go as well with the simple cell counts as with the correspondence analysis. 

To this referee who is not a scholar of day surgeries, their review of the literature seems adequate.  But it is presented as one paragraph, despite several logical breaks in the arguments. 

While I am quite a fan of graphical representations of results, I was quite disappointed by the use of the neutral terms, Dimension 1 and Dimension 2 as axes labels.  The authors need to explain what the close clustering and/or distance from the horizontal or vertical axes mean, and discuss what differences have what policy significance."

Interpretation of plots is similar to a case of factor analysis or of principal components analysis. Axes divide points by singular value decomposition of data matrix.

4."Line 215:  “Any reliable conclusions from single cell counts are impossible.”"

In our case there are cross tables with eight rows (regions) and with five columns (fields). It is not possible to orientate in 40 cell counts. It is not possible to say which region (field) is most similar (the closest) to any other region (field). Every region is point in five dimensional space of fields. And every field is point in eight dimensional space of regions. Correspondence analysis projects regions and fields to two dimensional plane with some information loss that can be controlled. Rows and columns of cross-table correspond. Viewpoint is symmetrical. Roles of cases and of variables are not predetermined. That is why the method is called correspondence analysis.

5. "Line 217 and 218:  “This is a reason why correspondence analysis was developed – to depict cross table.” is an inadequate rebuttal to value of Table 1."

See our other answers.

6. "Lines 223-227:  Similarly, this paragraph should be explaining what differences from the 1/5 th  row proportions contain more important information than the cell counts."

Correspondence analysis is based on proportions of counts than on counts alone. Otherwise, it would be principal components analysis.

7. "Line 242-243: “Inertia is measure of variance to explain the variance of cross table with corresponding dimensions.”  Please explain more clearly.  The use of dimensions appears to have a similar roll to either factors in Factor Analysis or to Eigen values in Eigen Vectors."

Yes you are right, we can roughly say that correspondence analysis is principal component analysis applied to cross table of counts with some distinctions like scaling etc. That is why, in SPSS tables and graphs, dimensions as approximation of original eight by five cross table are used.

8. Name of the paper is edited to the version "Application of Correspondence Analysis in Day Surgery System". We realise that output of this analysis does not create a comprehensive outcome, but it is a general analysis, so it is a sufficient name as it is abbreviated.